# Network-Based Approaches Reveal Potential Therapeutic Targets for Host-Directed Antileishmanial Therapy Driving Drug Repurposing

J. Eduardo Martinez-Hernandez,[a,b,c,h] Zaynab Hammoud,[d] Alessandra Mara de Sousa,[e] Frank Kramer,[d] Rubens L. do Monte-Neto,[e] Vinicius Maracaja-Coutinho,[c,f,h] Alberto J. M. Martin[b,g]

aPrograma de Doctorado en Genómica Integrativa, Vicerrectoría de Investigación, Universidad Mayor, Santiago, Chile

bLaboratorio de Biología de Redes, Centro de Genómica y Bioinformática, Facultad de Ciencias, Universidad Mayor, Santiago, Chile

cAdvanced Center for Chronic Diseases - ACCDiS, Facultad de Ciencias Químicas y Farmacéuticas, Universidad de Chile, Santiago, Chile

dIT-Infrastructure for Translational Medical Research, University of Augsburg, Augsburg, Germany

eBiotecnologia Aplicada ao Estudo de Patógenos - Instituto René Rachou – Fundação Oswaldo Cruz, Belo Horizonte, Minas Gerais, Brazil

fInstituto Vandique, João Pessoa, Brazil

gEscuela de Biotecnología, Facultad de Ciencias, Universidad Mayor, Santiago, Chile

hCentro de Modelamiento Molecular, Biofísica y Bioinformática – CM2B2, Facultad de Ciencias Químicas y Farmacéuticas, Universidad de Chile, Santiago, Chile

**ABSTRACT** *Leishmania* parasites are the causal agent of leishmaniasis, an endemic disease in more than 90 countries worldwide. Over the years, traditional approaches focused on the parasite when developing treatments against leishmaniasis. Despite numerous attempts, there is not yet a universal treatment, and those available have allowed for the appearance of resistance. Here, we propose and follow a host-directed approach that aims to overcome the current lack of treatment. Our approach identifies potential therapeutic targets in the host cell and proposes known drug interactions aiming to improve the immune response and to block the host machinery necessary for the survival of the parasite. We started analyzing transcription factor regulatory networks of macrophages infected with *Leishmania major*. Next, based on the regulatory dynamics of the infection and available gene expression profiles, we selected potential therapeutic target proteins. The function of these proteins was then analyzed following a multilayered network scheme in which we combined information on metabolic pathways with known drugs that have a direct connection with the activity carried out by these proteins. Using our approach, we were able to identify five host protein-coding gene products that are potential therapeutic targets for treating leishmaniasis. Moreover, from the 11 drugs known to interact with the function performed by these proteins, 3 have already been tested against this parasite, verifying in this way our novel methodology. More importantly, the remaining eight drugs previously employed to treat other diseases, remain as promising yet-untested antileishmanial therapies.

**IMPORTANCE** This work opens a new path to fight parasites by targeting host molecular functions by repurposing available and approved drugs. We created a novel approach to identify key proteins involved in any biological process by combining gene regulatory networks and expression profiles. Once proteins have been selected, our approach employs a multilayered network methodology that relates proteins to functions to drugs that alter these functions. By applying our novel approach to macrophages during the *Leishmania* infection process, we both validated our work and found eight drugs already approved for use in humans that to the best of our knowledge were never employed to treat leishmaniasis, rendering our work as a new tool in the box available to the scientific community fighting parasites.

Address correspondence to Vinicius Maracaja-Coutinho, vinicius.maracaja@uchile.cl, or Alberto J. M. Martin, alberto.martin@umayor.cl.

**KEYWORDS** drug repurposing, gene regulatory networks, host-direct therapy, leishmaniasis, multilayered network

Leishmaniases are a group of vector-borne neglected tropical diseases caused by *Leishmania* parasites (1). The clinical manifestations range from self-healing skin ulceration (cutaneous leishmaniasis [CL]) to splenomegaly and hepatomegaly (visceral leishmaniasis [VL], which is the deadliest form of leishmaniasis) (2). According to the last report for leishmaniasis emitted by the World Health Organization (WHO; www.who.int/en/news-room/fact-sheets/detail/leishmaniasis), it is estimated that annually there are about 1 million infections for CL and 90,000 for VL. Despite the availability of some drugs for leishmaniasis treatment (3), drawbacks in their current clinical use have been documented, ranging from high costs, toxicity, and the selection of resistant parasites (4).

Host-directed therapies (HDTs) are a group of strategies that interfere in the host mechanisms that are necessary for pathogen survival and/or stimulate the immune response to respond to pathogens and eliminate them, bypassing existing limitations with conventional treatments, such as the chance of developing resistance (5, 6). Recently, HDTs have been proposed for the treatment of diverse bacterial, viral, and parasitic diseases (7), such as tuberculosis (8), malaria (9), HIV infections (10, 11), and most recently COVID-19, caused by SARS-CoV-2 (12, 13).

To this day, a variety of strategies have been developed to identify new host-directed therapies for leishmaniasis treatment (5, 14). Many of these strategies are focused on the improvement of the immune response of the host (14). In previous work, Murray and colleagues demonstrated that a combination of interleukin 12 (IL-12) and the typical treatment with pentavalent antimony ($Sb^v$) helped in the recovery of animals infected with *Leishmania donovani*, proving that a joint therapy between drugs that improved the host's immune response and conventional therapies can be useful for the parasite elimination (15). Another study revealed that imatinib, an anticancer drug, was useful for reducing the severity of lesions caused by *Leishmania amazonensis* (16). Other studies were focused on promoting the production of interferon gamma (IFN-$\gamma$) (17), nitric oxide (NO) and interleukin 12 (IL-12) (18), and reactive oxygen species (ROS) (19), resulting in the improvement of immune response and promoting healing.

The use of network-based approaches to determine nonobvious biological interactions and their relationship to disease has recently increased with the continuous evolution of high-throughput technologies for transcriptomics, proteomics, and metabolomics assays, as well as the generated data available in public databases (20). These biological networks are a form of knowledge representation, used to structure different levels of relationship between bioentities (21). Networks can simplify the complexity and heterogeneity of biological systems and contain a myriad of knowledge eager to be explored through different specific computational approaches. Examples of this complexity are the Gene Regulatory Networks (GRNs), which represent regulatory events between regulatory elements, such as those between transcription factors (TFs) or noncoding RNAs (ncRNAs) and protein-coding genes (22). GRNs in combination with expression data can be used to infer condition-specific networks, which can be compared with control contexts or steady states, allowing the identification of possible disease markers or potential targets for a pharmacological treatment (23). However, biological networks represent not only regulatory interactions but also interactions connected by different means, such as nondirected protein-protein interactions or directed metabolic reactions. These aspects are employed in many fields of biomedicine by combining different networks, such as protein-protein interactions (PPIs), GRNs, and epidemiology information (24). The integration of the information available in the multitude of networks that one can obtain from a biological system is a complex task. To facilitate this process, multilayered graphs emerge as a solution recently introduced in the field of biomedicine (24). This concept consists of layering the components of the network, nodes, or edges, grouping them by their type based on the

heterogeneity of the network nodes or the relations among them. So, network elements of the same type are organized in the same layer and layers are connected by edges linking different types of elements (25).

Here, we present a combined approach to determine potential therapeutic targets for host-directed antileishmanial therapies. First, we employed GRNs based on transcriptomic data to model *Leishmania* infection dynamics in human macrophages, which are followed by a multilayered networks approach to map metabolic/signaling pathways and drug-target identification. Through this approach, we identified five potential novel targets according to their direct connection with 11 known drugs that could be repurposed to be used in host-directed antileishmanial therapy.

## RESULTS

**Global expression patterns in *Leishmania*-infected macrophages.** A global gene expression analysis was performed using a publicly available set of transcriptome sequencing (RNA-seq) data comprising four time points (4, 24, 48, 72 hours postinfection [hpi]) of *Leishmania major*-infected and non infected human macrophages previously reported by Fernandes et al. (26). This analysis revealed a high number of differentially expressed genes (DEGs) in paired-sample analysis (non infected against infected macrophage) at the first time points after infection. At 4 hpi, we observed a higher number of DEGs, totaling 4,704. At this time, 2,518 were upregulated and 2,186 downregulated in the infected macrophages compared to control non infected macrophages. We observed that DEGs decreased as the time postinfection increased, with the lowest number of DEGs at 72 hpi, 950, of which 411 were downregulated and 539 were upregulated (Fig. 1A).

Although the RNA-seq data correspond to a library obtained from poly A tailing RNAs, we identified the presence of a high number of differentially expressed ncRNAs. Most of these ncRNAs correspond to long noncoding RNAs (lncRNAs). We found at 4 hpi the highest number of overexpressed ncRNAs, with 428 upregulated (Fig. 1A), of which 407 were annotated as lncRNAs, 8 microRNAs (miRNAs), 6 miscellaneous RNAs (miscRNAs), 5 small nucleolar RNAs (snoRNA), and 1 small nuclear RNA (snRNA) (Table S1). On the other hand, the lowest number of up- (35) and downregulated (112) ncRNAs were identified at 72 hpi (Fig. 1A; Table S1). Interestingly, we observed 15 upregulated lncRNAs that are shared in all four time points (Fig. 1C, upper right; Data set S1). On the other hand, 20 downregulated lncRNAs were shared in the four time points (Fig. 1C, upper left; Data set S1). However, more analysis must be performed to determine the biological role of these ncRNAs in the host's response to *Leishmania* infection.

As with lncRNAs, we evaluated the number of protein-coding genes differentially expressed, their relationship with the immune system, and the GO term enrichment at the four time points postinfection. We identified between 470 (72 hpi) to 1,890 (4 hpi) upregulated protein-coding genes (Fig. 1B), of which 184 to 949 were related to immune system GO groups according to the ShinyGO analysis (27) (Data set S2), and 66 genes related to immune system were constantly upregulated in all four evaluated time points (Fig. 1C, bottom right; Data set S1). Here, we identify genes such as *JUN* or *MYC*, which have been reported as overexpressed in visceral leishmaniasis patients (28). As we show in Fig. 1B, the number of downregulated protein-coding genes ranges from 266 (72 hpi) to 1,768 (4 hpi). Our analysis of functional groups related to the immune system reveals the presence of several proteins that range from 90 (72 hpi) to 643 (4 hpi). Interestingly, we identified only 17 downregulated genes in all four time points (Fig. 1C, left bottom; Data set S1).

We applied a GO enrichment analysis to identify biological processes associated with host-pathogen interaction, response to stress, and immune response. Enriched GO categories for biological processes obtained in upregulated DEGs from infected against non infected macrophages comparison are listed in Data set S3. Genes upregulated at 4 hpi were enriched in GO categories involved in cytokine responses, such as

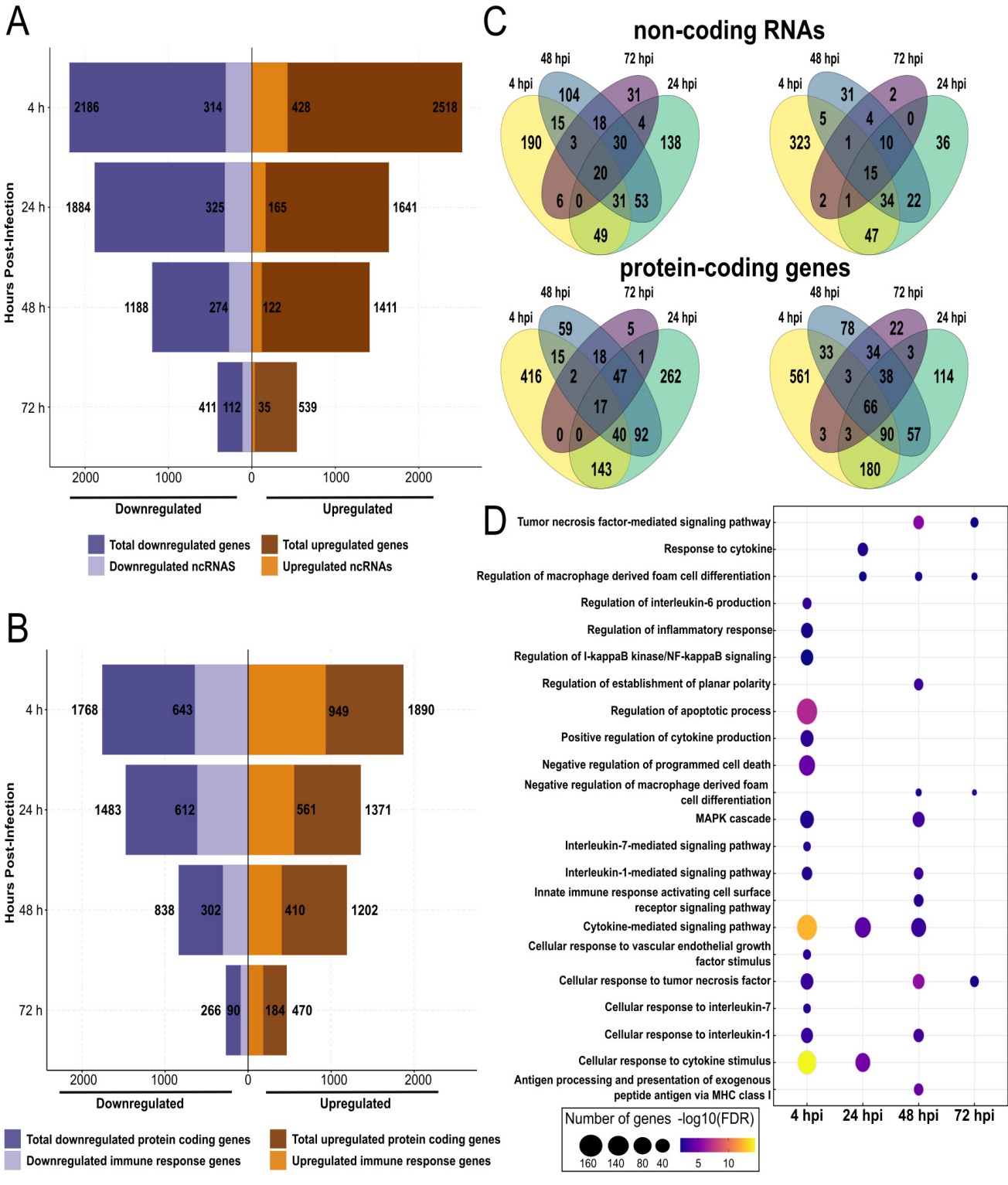

**FIG 1** Global transcriptomic profiles of *Leishmania*-infected human macrophages and genes related to immune response and host-pathogen interaction. Distribution of DEGs between different specific times postinfection. The box width indicates the number of DEGs downregulated (purple) and upregulated (orange) at adjusted *P* value of 0.05 and −0.5 > logFC > 0.5. Numbers at the end of each bar correspond to total DEGs obtained after paired-samples analysis. (A) Distribution of ncRNAs differentially expressed in *Leishmania major*-infected macrophages. (B) Distribution of protein-coding genes differentially expressed in *Leishmania major*-infected macrophages. (C) Venn diagrams exploring the conservation of ncRNAs (top) and protein-coding genes related to the immune system (bottom) in *Leishmania major*-infected macrophages. (D) Top 20 biological process GO terms enrichment related to immune response, stress, or host-pathogen interaction.

cellular response to cytokine stimulus (GO:0071345), cellular response to interleukin-1 (GO:0071347), an interleukin which promotes Th1 differentiation and inhibits disease progression in *Leishmania major* infections (29), and cellular response to interleukin-7 (GO:0071347), which enhances the elimination of amastigotes (30). On the other hand, terms related to the homeostasis of metals involved in macrophage function, such as zinc ion homeostasis (GO:0055069) (31), were also enriched. Mitogen-activated protein kinase (MAPK) cascade (GO:0000165), which is a determinant for IL-10 production and host susceptibility in *Leishmania* infections (32), was enriched in our GO analysis. Moreover, other GO terms, such as regulation of I-kappaB kinase/NF-$\kappa$B signaling (GO:0043122) and cellular response to tumor necrosis factor (GO:0071356), were enriched in upregulated genes in infected macrophages at 4 hpi (Fig. 1D).

Similar to that in 4 hpi, in 24 hpi timepoint we identified that cellular response to cytokine stimulus (GO:0071345) and cytokine-mediated signaling pathway (GO:0019221) terms were enriched within upregulated DEGs at this time point (Data set S3; Fig. 1D). Interestingly, we noticed the presence of enriched terms related to the regulation of macrophage differentiation at 24, 48, and 72 hpi (Fig. 1D; Data set S3). Antigen processing and presentation of exogenous peptide antigen via major histocompatibility complex (MHC) class I (GO:0042590) and Wnt signaling pathway, planar cell polarity pathway (GO:0060071), a pathway that canonically was related to defense against *Leishmania* infections (33), were enriched at 48 hpi but not at other time points (Fig. 1D). Also at 48 hpi, we identified terms such as cytokine-mediated signaling pathway (GO:0019221) and regulation of transcription from RNA polymerase II promoter in response to stress (GO:0043618) (Data set S3). Finally, our enrichment analysis showed that tumor necrosis factor-mediated signaling pathway (GO:0033209) and regulation of macrophage-derived foam cell differentiation (GO:0010743) were enriched at 72 hpi (Fig. 1D; Data set S3). Additionally, terms related to noncoding RNA (ncRNA) metabolism, such as ncRNA processing (GO:0034470), were consistently enriched in DEGs from the 24 and 48 hpi time points (Data set S3).

**Context-specific gene regulatory networks of *Leishmania*-infected macrophages reveal potential new therapeutic targets.** The TF-gene reference human GRN obtained after filtering high-quality connections is described in Table S2. After that, we used the normalized expression values obtained from our previous analysis to filter this gold standard GRN. We obtained eight time-specific GRNs (four to infected and four to non infected macrophages, one for each time point). Each context-specific network presented a different number of nodes and connections, as described in Table S2. The larger network for infected macrophages was obtained at 4 hpi, and it includes 19,750 nodes and 343,072 connections with 990 TFs. In addition, the smaller network of infected macrophages corresponds to 72 hpi and is made up of 19,718 nodes, of which 974 were identified as TFs, and 339,390 connections.

We compared each infected macrophage network against the control non infected macrophage networks obtained from the same time point. We found that non-TF genes show few alterations in the GRNs that represent macrophages in the first hours after infection, as indicated by F1 metric calculated by LoTo [90] for the presence or absence of network motifs in each compared GRN between 0.96 and 0.98. We also analyzed all TFs that had a significant change of regulation according to F1 values; for this, we used an F1 cutoff of 0.95 (Table 1). These results indicate that many TFs related to immune response show alterations in their regulations; in addition, not all these TFs were identified as differentially expressed according to our analysis. Therefore, we selected all nodes identified as differentially expressed and their connections present in the infected context networks and absent in non infected macrophage networks (Fig. 2A; Fig. S1A; Fig. S2A; Fig. S3A). Figure 2B shows the subnetwork after filtering all the edges present only in the infected macrophage network at 4 hpi. In this subnetwork, there are 160 connections and 63 nodes, of which 11 were TFs. Additionally, as listed in Table S3, we found 244 nodes for 24 hpi network comparison (Fig. S1B), 155 for 48 hpi (Fig. S2B), and 52 for 72 hpi (Fig. S3B).

To select a list of initial candidates, we evaluated all genes recovered from these comparisons to remove all non-DEGs, as well as genes not related to immune

**TABLE 1** Transcription factors (TF) with higher changes in their regulations in *Leishmania major*-infected macrophages[a]

| Time point | TF | F1 | Function |
|---|---|---|---|
| 4 h | MEF2B | 0 | |
| | PROX1[b] | 0 | Regulation of developmental process |
| | KLF1[b] | 0.339 | Immune system process |
| | E2F2[b] | 0.611 | Regulation of developmental process |
| | FLI1[b] | 0.931 | Immune system process |
| | STAT4[c] | 0.934 | |
| | TCF3 | 0.935 | Immune system process/leukocyte activation |
| 24 h | MEF2B | 0 | |
| | NFE2 | 0 | Immune system process |
| | PROX1[b] | 0 | Regulation of developmental process |
| | POU5F1 | 0.019 | Response to stress |
| | TCF7[b] | 0.078 | Immune system process/leukocyte activation |
| | ATF6 | 0.536 | Response to stress |
| | CEBPD[b] | 0.850 | Immune system process |
| | NCOA2[b] | 0,853 | Response to endogenous stimulus |
| | TCF3 | 0.899 | Immune system process/leukocyte activation |
| | ZEB1 | 0.907 | Immune system process/leukocyte activation |
| | RUNX3 | 0.918 | Immune system process/leukocyte activation |
| | FLI1[b] | 0.930 | Immune system process |
| | FOXO3 | 0.937 | Immune system process |
| | EPAS1 | 0.949 | Immune system process |
| 48 h | MEF2B | 0 | |
| | POU5F1 | 0.019 | Response to stress |
| | TBX21 | 0.245 | Immune system process/leukocyte activation |
| | SOX6 | 0.412 | Immune system process |
| | TP73[b] | 0.478 | Immune system process |
| | CEBPD[b] | 0.854 | Immune system process |
| | FLI1[b] | 0.929 | Immune system process |
| | ZEB2 | 0.930 | Response to stress |
| | STAT4 | 0.935 | |
| | TCF3 | 0.935 | Immune system process/leukocyte activation |
| | TCF4 | 0.942 | Regulation of response to stimulus |
| 72 h | MEF2B | 0 | |
| | POU5F1 | 0.019 | Response to stress |
| | TCF7[b] | 0.021 | Immune system process/leukocyte activation |
| | TBX21 | 0.245 | Immune system process/leukocyte activation |
| | ELF3 | 0.335 | Response to stress |
| | SOX6 | 0.362 | Immune system process |
| | CEBPD | 0.849 | Immune system process |
| | RUNX3 | 0.912 | Immune system process/leukocyte activation |
| | IKZF1 | 0.917 | Immune system process/leukocyte activation |
| | ZEB2 | 0.932 | Response to stress |
| | STAT4 | 0.934 | |
| | TCF3 | 0.936 | Immune system process/leukocyte activation |

[a]TF, transcription factor; F1 represents the harmonic mean between precision and recall, ranging from 0 to 1, in which 1 represents a higher similarity of node *X* in both networks.
[b]Downregulated gene.
[c]Upregulated gene.

response, host-pathogen interaction, and/or stress. This first selection contains a total of 373 genes, of which only 113 passed all filters. Interestingly, we identified several TFs that act in the activation of the immune response, such as *JUN*, or the negative regulation of immune response, such as *MYC*. We also determined with this analysis that some effectors of the immune response, such as IL-16 (a proinflammatory interleukin), were present in our list of possible therapeutic targets. In Data set S4, we summarize all the identities, expression values, and functions of all 113 candidate genes.

Finally, we used the 113 selected genes as seeds to obtain the biological pathways in which their coded product participates. We obtained a list of 313 Reactome IDs of

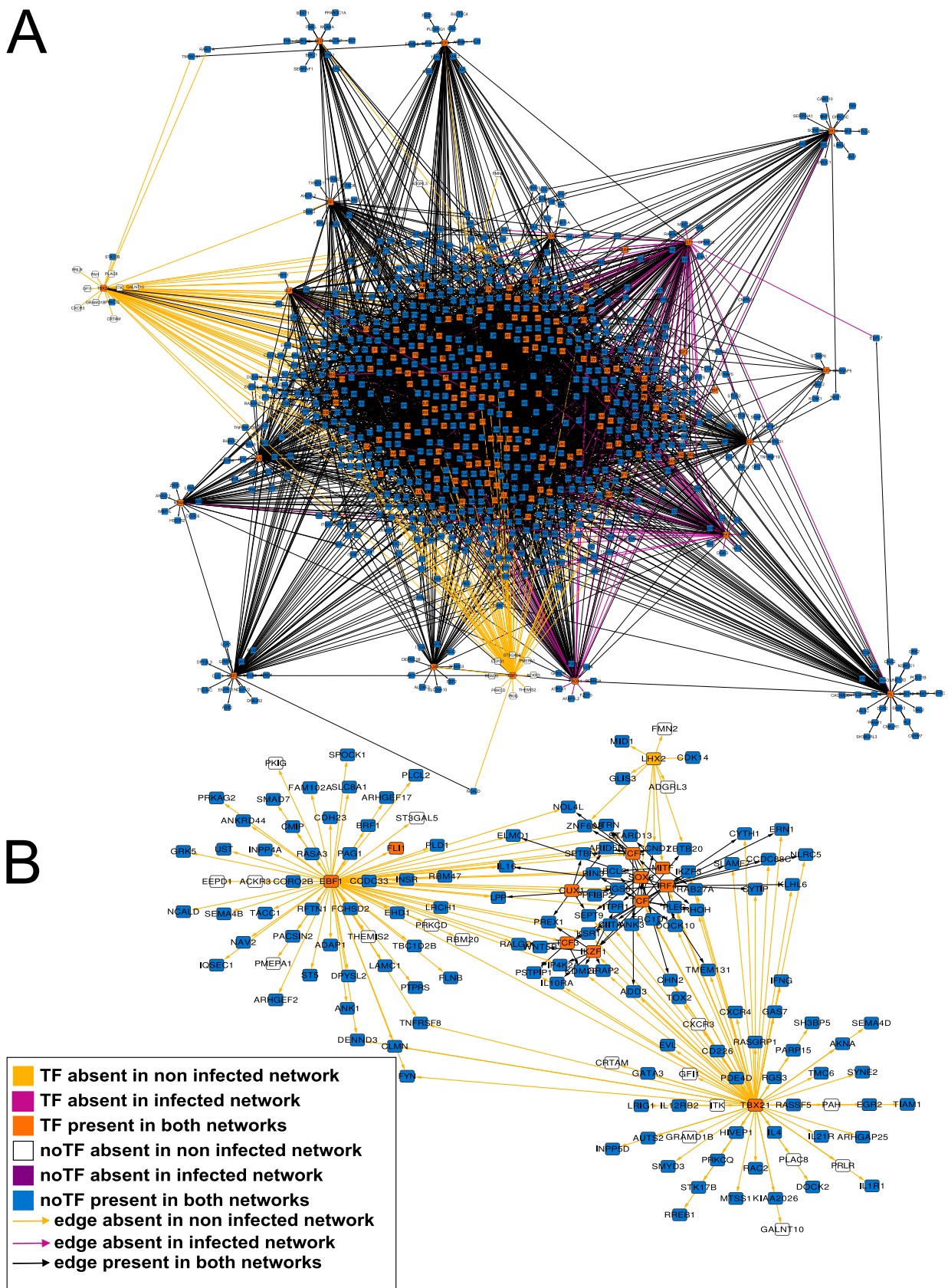

**FIG 2** Network comparison of non infected against infected macrophage at 4 h postinfection. (A) The network shown is formed by 942 nodes (167 TFs) and 3,847 edges colored according to their existence in the non infected macrophage network, infected-macrophage network, or both networks. (B) Subnetwork represents all edges presented only in the 4 hpi network. The colors of edges and nodes are the same as those in the upper network.

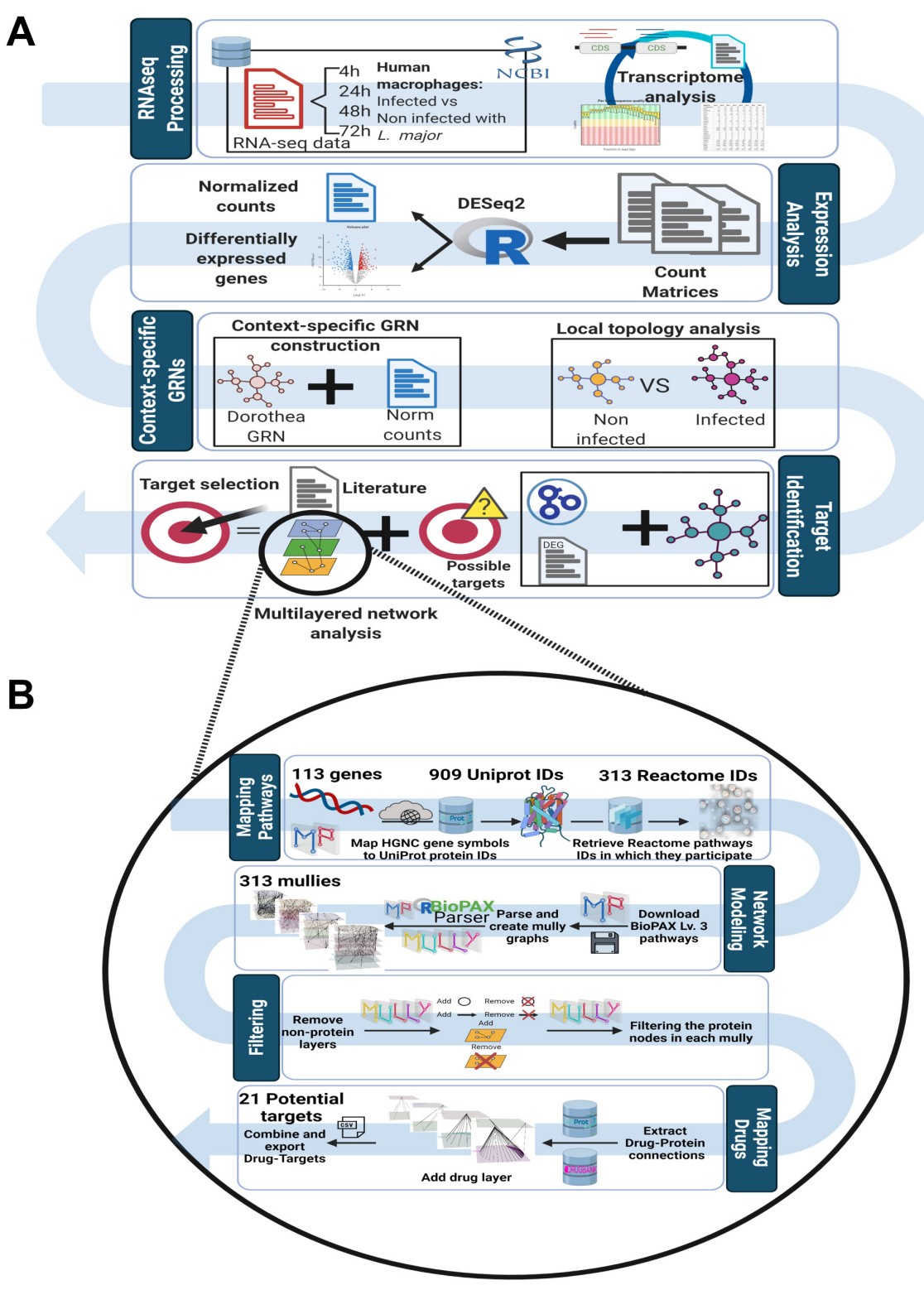

**FIG 3** Pipeline to identify potential therapeutic targets for leishmaniasis host-directed treatment in human macrophage from RNA-seq data. (A) First, we processed a set of RNA-seq data derived from *Leishmania major*-infected macrophages. This data set is composed of 4 time points: 4 h postinfection (hpi), 24 hpi, 48 hpi, and 72 hpi. Raw reads were analyzed using an in-house-developed pipeline that takes raw reads as input, and as output we obtained bona fide read counts per gene. Then, counts were used to obtain a normalized counts matrix and detect the differentially expressed genes. Next, we filtered a reference human GRN using normalized data to contextualize the GRN and get infected and non infected contexts simultaneously. After that, we applied a

Reactome pathway entries linked to 909 UniProt IDs, which we downloaded and used to generate 313 mully multilayered graphs. After mapping the protein layers to our list of selected candidates, the mully graphs were filtered by deleting all nonprotein layers while adding the transitive edges to preserve the connections. Then, we deleted all protein nodes that were present in the mully graphs but not included in our list of candidates. Next, from these filtered graphs, we extracted the drug connections from UniProt IDs of our selected list and DrugBank, adding a drug layer to each filtered pathway graph. We combined all drug-protein connections from the different graphs to obtain the final list of drug targets (Fig. 3B). A final set of 21 gene-pathway-drug interactions were obtained. In total, we identified 124 different biological pathways and 331 different drugs that have a direct connection with these 21 genes.

To reduce the number of drug-target interactions, we filtered this list using as selection criteria those drugs that were annotated as "approved," as well as "approved-investigational" or "approved-vet_approved" combinations. A total of 195 approved drugs were finally selected and presented direct interaction with 13 different genes. After this selection, we evaluated these 13 genes to identify their relationship with *Leishmania* infection. We found 8 genes that were confirmed in the literature as genes involved in *Leishmania* infection (34–38). In Table S4, we included the list of 8 genes that are potential candidates as host-directed therapeutic targets. We found that these 8 genes have a direct connection to 145 different drugs. This information was cross-linked with the expression data and other related metadata, such as drug-target interaction, evidence of the previous usage as antileishmanial drugs, current usage, or side effects, to determine the best set of drugs that could be repurposed as host-directed therapy for the treatment of leishmaniasis. After this analysis, we identified the coding product of five genes, androgen receptor (*AR*), C-Jun (*JUN*), platelet-derived growth factor receptor alpha (*PDGFRA*), prostaglandin endoperoxide synthase 2 (*PTGS2*), and vascular endothelial growth factor A (*VEGFA*), as potential therapeutic targets. In the pathway analysis, we identified that the product of these five genes has direct participation in pathways such as detoxification of reactive oxygen species, regulation of the apoptosome activity (*JUN*), or SCF/c-kit signaling pathway (*PDGFRA*) (Table S4). Furthermore, 11 drugs were selected by their direct interaction with the products of these five genes as well as their meta-features. Clascoterone and adapalene, which have previously been used as drugs to treat skin conditions such as acne, and other anti-inflammatories drugs, such as tolfenamic acid and flufenamic acid, were proposed as potential drugs for leishmaniasis chemotherapy. Moreover, the antipsychotic acetophenazine and the antineoplastic ripretinib were also selected. Table 2 displays more information about the potential targets, their interaction with drugs, their usage, and other relevant information. Our results indicated that several new therapeutic targets could be identified from the changes in gene regulation that occur during the infective process of *Leishmania major* of human macrophages, coupled with the integration of data related to pathways and drug-target direct connections.

## DISCUSSION

Conventional therapies to tackle leishmaniasis are typically composed of only a few different drugs (39, 40). Moreover, these therapies present several drawbacks, affecting mainly their efficiency, toxicity, and the ability to select resistant parasites (4). Recently, several approaches for the treatment of parasitic diseases were focused on host-directed therapies (5), aiming to repurpose drugs previously used for other diseases (41). This strategy search identifies new uses for drugs used or candidates in advanced clinical phases through network screening, followed by phase II and III

**FIG 3** Legend (Continued)

pairwise comparison of infected against non infected contextualized networks to obtain the nodes and connections present in a disease condition. Next, we used the list of nodes to keep only genes involved in processes related to immune response, response to stress, or host-pathogen interaction and that were evidenced as differentially expressed. (B) Schematic workflow was applied to identify the drug targets using the Multipath package. With the filtered list, we mapped the gene set of interest to their gene products and related biological pathways in which these proteins participate and obtained the drug-gene product direct connection. Finally, drug-target interactions were literature filtered to select the best candidate targets for host-directed antileishmanial treatment.

**TABLE 2** Potential therapeutic targets for host-directed leishmaniasis treatment and their best drug connection

| Target | Drug | DrugBank ID | Pharmacological action | Action | Route of administration | Use | Antileishmanial evidence |
|---|---|---|---|---|---|---|---|
| AR | Esculin | DB13155 | Hyaluronidase and collagenase inhibitor | Agonist | Oral route | Vasoprotective agent | Yes (46, 47) |
| | Clascoterone | DB12499 | Testosterone and dihydrotestosterone blocker | Antagonist | Topical application | Acne | No |
| | Acetophenazine | DB01063 | D2 and 5HT2 inhibitor | Antagonist | Oral route | Antipsychotic | No |
| JUN | Adapalene | DB00210 | AP-1 and TLR-2 inhibitor | Antagonist | Topical application | Acne | Yes (80) |
| PDGFRA | Ripretinib | DB14840 | PDGFRB, BRAF, VEGF, and TIE2 inhibitor | Inhibitor | Oral route | Anticancer | No |
| PTGS2 | Tolfenamic acid | DB09216 | COX inhibitor | Antagonist | Oral route | Nonsteroidal anti-inflammatory drug (NSAID) | No |
| | Flufenamic acid | DB02266 | COX inhibitor | Unknown | Oral route | Nonsteroidal anti-inflammatory drug (NSAID) | Yes (not effective) (99, 100) |
| | Antrafenine | DB01419 | COX inhibitor | Inhibitor | Oral route | Analgesic | No |
| VEGFA | Dalteparin | DB06779 | Factor Xa inactivator | Inhibitor | Intracutaneous injection | Thrombosis | No |
| | Minocycline | DB01017 | VEGFA inhibitor | Inhibitor | Oral and topical | Antibiotic | No |
| | Pidolic acid | DB03088 | Unknown | Unknown | Oral and topical | Moisturizer for dry skin | No |

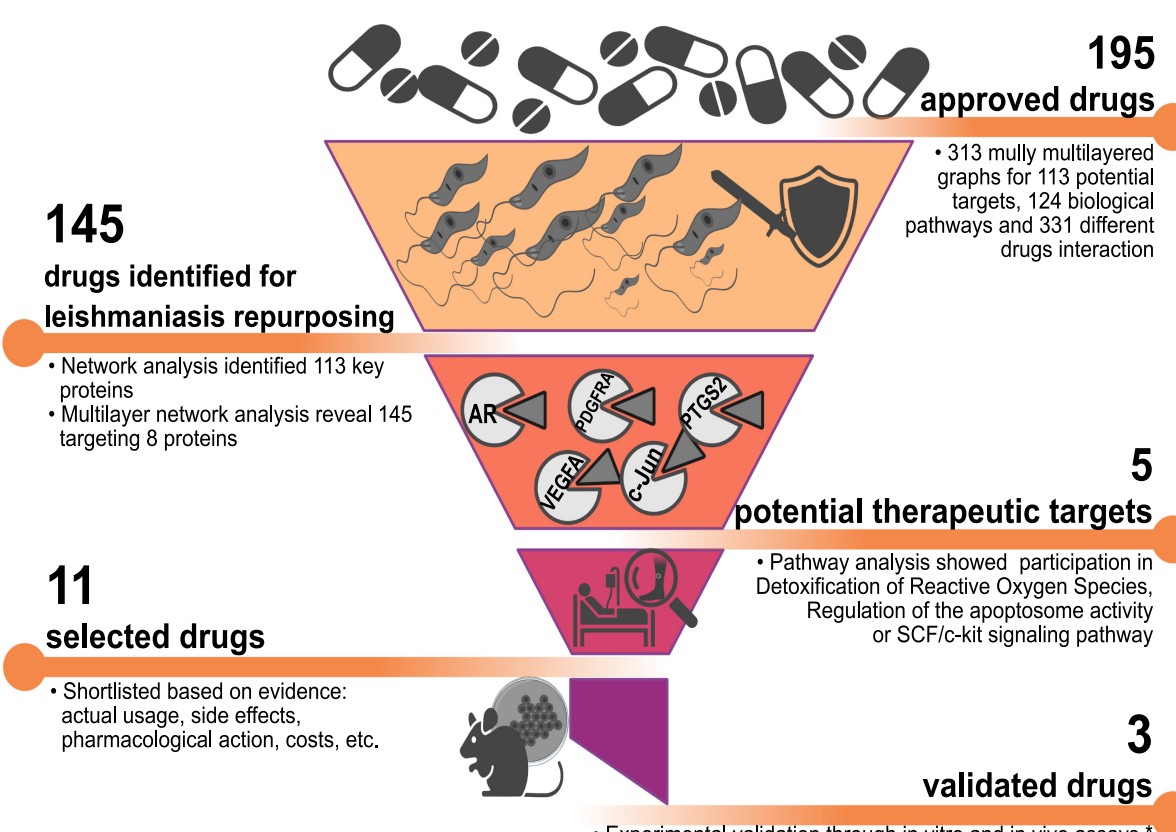

**195**
approved drugs
• 313 mully multilayered graphs for 113 potential targets, 124 biological pathways and 331 different drugs interaction

**145**
drugs identified for leishmaniasis repurposing
• Network analysis identified 113 key proteins
• Multilayer network analysis reveal 145 targeting 8 proteins

**5**
potential therapeutic targets
• Pathway analysis showed participation in Detoxification of Reactive Oxygen Species, Regulation of the apoptosome activity or SCF/c-kit signaling pathway

**11**
selected drugs
• Shortlisted based on evidence: actual usage, side effects, pharmacological action, costs, etc.

**3**
validated drugs
• Experimental validation through in vitro and in vivo assays *

**FIG 4** Context-specific gene regulatory networks of *Leishmania*-infected macrophage and multilayered network analysis reveal potential new therapeutic targets and drug repurposing for host-directed antileishmanial therapies. Our network analysis reveals a final set of 5 possible drug targets; these 5 targets interact with 11 different drugs. Our literature mining reveals that at least 3 drugs were validated in *in vitro* or *in vivo* models to test their potential as antileishmanial drugs (46, 47, 80, 99, 100).

clinical trials (42). Here, we employed network-based approaches to identify novel therapeutic strategies by determining drug target proteins that play key roles in the host during leishmaniasis. Our results demonstrate how network-based approaches allow for the effective identification of new therapeutic targets for host-directed treatment in parasitic diseases. Compared to the traditional process, drug repositioning has some advantages, in that its cost and development time are reduced.

In this study, we found five genes whose encoded products have the potential to be new therapeutic targets. We identified a direct connection between these gene products and 11 FDA-approved drugs (Table 2, Fig. 4). Many of these drugs have previously been used to treat skin disease (e.g., acne) as anti-inflammatories or antineoplastics.

Drugs such as clascoterone are antagonists of the androgen receptor (*AR*) for the treatment of androgen-dependent skin diseases, including androgenetic alopecia and acne. Previous reports have shown that *AR* has a relevant role in the immune response derived from parasitic infections (38). Two different works, reported by Sánchez-García and colleagues and Qiao and colleagues, discovered that hormones such as dihydro-testosterone (DHT) and testosterone that interact with the androgen receptor can alter parasites' development and survival or inhibit apoptosis in *Leishmania*-infected macrophages (43, 44). These effects seem to be exerted by specific receptors for androgen in macrophages, leading to greater replication of the parasite and an increased rate of infection, with increased proliferation of the parasite. Acetophenazine is a moderately potent antipsychotic and antagonist of the dopamine D2 receptor and the androgen receptor. Previous studies show that dopaminergic receptor antagonists can inhibit the growth and multiplication of toxoplasmosis parasites (45).

Esculin, a phenolic compound, typically used as a vasoprotective agent, was tested

to evaluate their inhibitory activity against *Leishmania infantum* arginase (ARGLi), demonstrating a null inhibitory activity in *in vitro* assay (46). On the other hand, this compound promotes a reduction of transmission and viability of *L. infantum* and *Leishmania mexicana* in a vector model assay (47). Notwithstanding that our findings showed *AR* as a possible target for leishmaniasis treatment, studies with tamoxifen, an anticancer drug that acts as a nonsteroidal estrogen receptor modulator, presents antileishmanial activity (48, 49) by inhibiting the parasite's inositol phosphorylceramide (50) with reduced chances of selecting resistant parasites (51). It is worth noting that tamoxifen treatment resulted in scrotal swelling that leads to infertility in cutaneous experimental leishmaniasis caused by *L. major* in male mice (52). Androgen is converted to estrogen by an aromatase enzyme that can be inhibited by testosterone (53). In humans, combined therapy using oral tamoxifen and meglumine antimoniate in patients suffering from cutaneous leishmaniasis resulted in cure rates similar to those of conventional schemes (54). Initially used to treat breast cancer, tamoxifen is a classic example of a repurposed drug with different targets in the parasite that was promising during *in vitro* investigations but presented limitations when following the drug development pipeline.

Regarding the viability of other potential targets in our list, we found that prostaglandin endoperoxide synthase 2 (*PTGS2*) was previously reported as a biomarker for the response to infections for *L. major* (34). Our results showed a consistent misregulation of this gene that is consistent with these previous studies. Also, we identified a great number of direct connections between *PTGS2* and different drugs, such as tolfenamic acid, a nonsteroidal anti-inflammatory drug (NSAID) usually used for migraine pain. Also, this drug shows antibacterial activity against *Staphylococcus aureus* (55), *Burkholderia pseudomallei* (56), *Liberibacter asiaticus* (57), and in lesser potency, *Vibrio cholerae* (58). Additionally, in a recent study, it is shown to be effective to kill schistosomes in *in vitro* and *in vivo* assays (59). In this regard, *Leishmania braziliensis* prostaglandin F2$\alpha$ synthase (LbrPFG2S) was associated with host-parasite interaction playing a crucial role in pathogenicity through proinflammatory lipid synthesis (60, 61), revealing and validating a potential intervention in a parasite's prostaglandin biosynthesis pathway.

Our approach also found different anticancer drugs among those targeting our list of candidates. This type of drug has been used before to treat leishmaniasis. For example, miltefosine was developed as an anticancer drug, but today it is a choice for visceral leishmaniasis treatment (62, 63). Ripretinib was described as a promising drug for the treatment of gastrointestinal stromal tumors, and more recently its drug family has been proposed to be repositioned for COVID-19 treatment (64). Our approach also indicated that ripretinib might be useful for the treatment of leishmaniasis. This is because ripretinib acts as a kinase inhibitor and presents an inhibition activity over that of platelet-derived growth factor receptor (*PDGFRA*), a tyrosine kinase receptor previously targeted for antileishmanial therapies that presents a significant reduction in parasitic survival (65). Indeed, the anticancer drug sunitinib, a broad-spectrum receptor tyrosine kinase inhibitor, blocked progressive splenomegaly and improved immunity as adjuvant therapy in murine experimental leishmaniasis (66). Flufenamic acid, a nonsteroidal anti-inflammatory, showed good antimalarial activity, significantly retarding the intraerythrocytic growth of *Plasmodium falciparum*, but its antileishmania potential still needs to be tested (67).

Another candidate target found by our method is vascular endothelial growth factor A (*VEGFA*), a gene whose expression is induced during *Leishmania major* infection (68, 69). The product of this gene promotes lymphangiogenesis, a process that is involved in the inflammatory response and lesion healing (68). The product encoded by *VEGFA* is used as a target in antiangiogenesis therapies to reduce the vascularization of tumors (70). Minocycline is an antibiotic from the tetracycline family that interacts with the VEGFA protein and inhibits angiogenesis (71). This antibiotic has been proposed to treat parasitic infection in the late 1980s, mainly against *Giardia lamblia*, due

to a potent activity that reduced the survival of the parasite *in vitro* (72). The use of minocycline in the treatment of malaria has shown a decrease in T-cell-mediated brain inflammation and a reduction in gene expression independent of the antiparasitic property (73). Nonetheless, a more recent study reported that a patient diagnosed with leishmaniasis was treated with minocycline (74). However, the treatment was stopped shortly after and nonconclusive results were obtained. Pidolic acid is an active form of 5-oxoproline and occurs in fundamental biological processes such as intracellular stage differentiation, host cell infection, and resistance to various stresses in protozoa belonging to the genera *Trypanosoma* and *Leishmania*. Fargnoli et al. showed in an unprecedented way that L-proline uptake has been proposed as a chemotherapeutic target for *Trypanosoma cruzi*, opening a new horizon in the development of new chemotherapeutics against Chagas disease and other parasitic diseases (75).

The last target in our list is *JUN*, a gene that encodes the c-Jun protein that is a central part of the AP-1 transcription factor. This transcription factor is crucial for the inactivation of macrophages during *Leishmania* infection (76). Adapalene is a retinoid that acts as a comedolytic and anti-inflammatory agent that has been widely used due to its milder effects compared to those of other substances of the same class, mainly used for acne treatment that targets c-*JUN* via the AP-1 transcription factor (77, 78). Interestingly, adapalene has been used as an anticancer drug in an *in vitro* assay promoting the apoptosis of colorectal cells (79). The antileishmanial activity of adapalene was tested *in vitro* and *in vivo* in animal models, showing high antileishmanial activity and promoting healing in hamsters infected with *Leishmania panamensis* (80). This drug not only affects the parasite but also promotes the host's immune response, showing that it could be a very effective drug for leishmaniasis treatment.

## CONCLUSION

We have created a new strategy for the search for therapeutic targets. We applied our approach to finding host-directed antileishmanial therapeutic targets and showed that it provides a significant number of novel potential targets. This new strategy helps to bypass common issues arising from conventional antiparasitic therapies, such as the fast appearance of resistance and strong side effects that preclude generalized drug usage. Furthermore, we demonstrate in this study that many drugs could be repositioned for leishmaniasis treatment. Importantly, all drugs selected in our work need to be experimentally tested to confirm their potential as host-directed antileishmanial therapies. Notably, after demonstrating the potential of our new strategy for the identification of therapeutic targets, it is important to highlight that it requires only transcriptomic data that is integrated with other available data, making it an easy tool to adapt for other diseases.

Our results indicate the possibility of repurposing several drugs that could be useful as antileishmanial therapies. However, even with strong *in silico* evidence supporting our list of new therapeutic targets, our approach still requires *in vivo* testing to confirm that these repurposed drugs would be useful as treatments for *Leishmania* infections.

## MATERIALS AND METHODS

**RNA-seq data sets from *Leishmania*-infected macrophages.** A set of 43 RNA sequencing (RNA-seq) samples derived from human macrophages infected with *Leishmania major* (BioProject accession number PRJNA290995) were downloaded from NCBI SRA (81). This data set is composed of 4 time points, 4 h postinfection (hpi), 24 hpi, 48 hpi, and 72 hpi, five biological replicates per time point, and six biological replicates for uninfected macrophages as a control at the same time points as infected macrophages.

**RNA-seq data analysis.** RNA-seq raw data quality control inspection was performed using FastQC version 0.11.8 (https://www.bioinformatics.babraham.ac.uk/projects/fastqc/). Low-quality reads were removed with Trimmomatic version 0.36 (82) using a Phred cutoff value (Q) of 30. Then, we mapped the remaining high-quality reads with Hisat2 version 2.1.0 (83) to the human genome (GRCh38), downloaded from Ensembl (84). Expression values were calculated using HTSeq-count version 0.7.2 (85). The resulting counting reads matrix was normalized and used to identify the differentially expressed genes through DESeq2 version 1.24.0 (86). Genes with an adjusted $P$ value of $\leq 0.5$ and absolute log fold change of $\geq 0.5$ were considered differentially expressed.

**Gene regulatory network analyses.** A human reference GRN was obtained from DoRothEA release 1.3.3 (87), considering only high-quality connections (A, B, and C evidence codes reported in their original work). This reference GRN was then filtered using RNA-seq normalized data in the same way as that described by Santander et al. (88), with some modifications related to the expression threshold employed to contextualize the GRN. Here, interactions between a transcription factor (TF) and its target gene were saved only if DESeq2 normalized expression values (median ratios) for that TF were at least 10 (89). Then, we obtained contextualized networks for each condition (8 GRNs, 4, 24, 48, and 72 hpi, infected and non infected human macrophages) and applied a pairwise comparison of infected against non infected for all four serial times using LoTo (90). LoTo identifies genes whose regulatory environment varies based on binary classification metrics calculated with the presence or absence of network motifs in each compared GRN, determining differentially regulated nodes. Once these comparisons were obtained, we evaluated the F1 values of all TF and non-TF genes. F1 represents the harmonic mean between precision and recall, ranging from 0 to 1, in which 1 represents a higher similarity of node $X$ in both networks (90). Next, we crossed the list of differentially regulated genes in each comparison with the list of differentially expressed genes from DESeq2 analysis. Finally, gene candidates with F1 lower than 0.95 and an absolute value of logFC greater than 0.5 were considered in further analyses. The resulting list was functionally enriched using EnrichR version 3.0 (91–93) and ShinyGO v0.61 (27) and then filtered to keep only genes associated with processes related to immune response, response to stress, or host-pathogen interaction.

**Gene-protein, protein-pathway, and drug-target interaction mapping.** Multipath version 1.0.3 is an R package used to generate integrated reproducible pathway knowledge (94). Using Multipath, BioPAX-encoded pathways (95) can be modeled into multilayered graphs, where the biological pathways components are embedded into different layers based on their biological type. The built graphs are reproducible, i.e., all modifications applied to the graphs are stored. Multipath is also used to integrate influencing pathway knowledge from external databases like drugs from DrugBank version 5.1.8 (96). We used this package to query pathway knowledge databases and fetch relevant information needed in our computational analysis. To map the gene set of interest to their gene products from UniProt, Multipath uses UniProt.ws to fetch the UniProt IDs of the corresponding proteins, which were mapped to the list of candidates. Then, we got a list of biological pathways from Reactome version 73 Released (97) in which these proteins participate. These Reactome IDs were downloaded to generate mully multilayered graphs (98). Next, we filtered all proteins that were not coded by genes in our previous list. Finally, we extracted the drug targets from UniProt release 2020_05 and DrugBank version 5.1.8 and added a drug layer to each filtered pathway graph, preserving only those genes for which drug direct connection was identified (Fig. 3B).

**Selection of potential host-directed therapeutic targets for leishmaniasis treatment.** After drug-target direct interactions were obtained, we filtered this network to reduce the number of potential targets. To do so, we first selected those drug-target connections in which the drug had at least one of the following labels: approved, approved and investigational, or approved and vet_approved. After that, literature mining was applied to identify all genes that were playing a role in *Leishmania* infection and were identified as differentially expressed in infected macrophages in the transcriptomic analysis. Next, we used this information to refine the filtering of selected drug-target interactions. Finally, we evaluated all filtered drugs to obtain the type of drug-target interaction, evidence of previous usage as antileishmanial drugs, actual usage, side effects, pharmacological action, and any other relevant information for the best target selection. Figure 3 summarizes the entry pipeline employed in this work.

## SUPPLEMENTAL MATERIAL

Supplemental material is available online only.

**SUPPLEMENTAL FILE 1**, XLSX file, 0.05 MB.
**SUPPLEMENTAL FILE 2**, XLSX file, 0.05 MB.
**SUPPLEMENTAL FILE 3**, XLSX file, 0.1 MB.
**SUPPLEMENTAL FILE 4**, XLSX file, 0.02 MB.
**SUPPLEMENTAL FILE 5**, PDF file, 0.1 MB.
**SUPPLEMENTAL FILE 6**, PDF file, 7.6 MB.

## ACKNOWLEDGMENTS

This project was funded by FONDECYT regular 1181089 to A.J.M.M. and FONDAP 15130011 and FONDECYT regular 1211731 to V.M.C., from Agencia Nacional de Investigación Científica y Desarrollo (ANID), by Concurso Apoyo a la Infraestructura para la Investigación 2019 (INFRA-021/01/2019), Vicerrectoría de Investigación y Desarrollo, Universidad de Chile to V.M.C., by a PhD scholarship from Universidad Mayor to J.E.M.H., and by German Ministry of Education and Research (Bundesministerium für Bildung und Forschung, BMBF) grants, respectively, FKZ01ZX1508 and FK01ZX1708D. A.M.S. holds a PhD fellowship from Coordenação de Aperfeiçoamento de Pessoal de Nível Superior - Capes, Proex no. 88887.604141/2021-00, Brasil. R.M.N. is a CNPq research

fellow (312965/2020-6). The funders had no role in the design of the study, collection, analysis, and interpretation of data, or writing the manuscript.

Powered@NLHPC: this research was partially supported by the supercomputing infrastructure of the NLHPC (ECM- 02); and by the computing infrastructure of the Centro de Genómica y Bioinformática, Universidad Mayor.

We declare no conflicts of interest.

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
