## [Reviewer comments · Microbiology Spectrum]

**Microbiology
Spectrum**

Network-based approaches reveal potential therapeutic targets for host-directed antileishmanial therapy driving drug repurposing

J. Eduardo Martinez-Hernandez, Zaynab Hammoud, Alessandra Mara de Sousa, Frank Kramer, Rubens Monte-Neto, Vinicius Maracaja-Coutinho, and Alberto Martin

Corresponding Author(s): Alberto Martin, Universidad Mayor

Review Timeline:

Submission Date:

July 26, 2021

Accepted:

August 27, 2021

Editor: Tim Downing

Reviewer(s): The reviewers have opted to remain anonymous.

Transaction Report:

DOI: <https://doi.org/10.1128/Spectrum.01018-21>

August 27, 2021

Dr. Alberto J. M. Martin
Universidad Mayor
Centro de Genómica y Bioinformática
Santiago
Chile

Re: Spectrum01018-21 (Network-based approaches reveal potential therapeutic targets for host-directed antileishmanial therapy driving drug repurposing)

Dear Dr. Alberto J. M. Martin:

Your manuscript has been accepted, and I am forwarding it to the ASM Journals Department for publication. You will be notified when your proofs are ready to be viewed.

Sincerely,

Tim Downing
Editor, Microbiology Spectrum

Journals Department
Fig S3: Accept

Data Set 2: Accept
Supplementary Tables: Accept
Data Set 3: Accept
Fig S1: Accept
Fig S2: Accept
Data Set 1: Accept
Data Set 4: Accept